# Synthesis and Catalytic Activity of Activated Carbon Supported Sulfonated Cobalt Phthalocyanine in the Preparation of Dimethyl Disulfide

**Zhiliang Cheng [1,2], Mingxing Dai [2], Xuejun Quan [1,2,*], Shuo Li [2,*], Daomin Zheng [1], Yaling Liu [1] and Rujie Yao [1]**

[1] Chongqing Unis Chemical Company Ltd., Chongqing 401121, China; purper886@126.com (Z.C.); zhengdaomin@163.com (D.Z.); liuyaling886@126.com (Y.L.); ccyao@163.com (R.Y.)
[2] School of Chemistry and Chemical Engineering, Chongqing University of Technology, Chongqing 400054, China; dai_mingxing@yeah.net
* Correspondence: hengjunq@cqut.edu.cn (X.Q.); lishuo@cqut.edu.cn (S.L.); Tel.: +86-(0)23-6256-3180 (X.Q.); +86-(0)23-6256-3183 (S.L.)

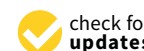

**Featured Application: Dimethyl disulfide (DMDS) is an important fine chemical, which is widely used as soil fumigant, herbicide, food additive, etc. and exhibits broad market prospects. In this work, an activated carbon (AC)-supported sulfonated cobalt phthalocyanine (AC-CoPcS) catalyst through a chemical linkage of ethylenediamine was successfully synthesized and applied to the preparation of DMDS. The new catalyst shows better catalytic and reuse performance than the commercial one, and exhibits good potential for the industrial application of manufacturing DMDS.**

**Abstract:** The Merox process was widely applied in the fine chemical industry to convert mercaptans into disulfides by oxidation with oxygen, including dimethyl disulfide (DMDS). In this paper, a new activated carbon (AC)-supported sulfonated cobalt phthalocyanine (AC-CoPcS) catalyst was prepared through the chemical linkage of ethylenediamine between them. UV−VIS, FT-IR, BET, and XPS were used to characterize the structure of the new catalyst. Then AC-CoPcS was applied to catalyze sodium methylmercaptide (SMM) oxidation for the preparation of DMDS. The effect of process parameters, such as reaction time, catalyst dosage, reaction temperature, and oxygen pressure on SMM conversion per pass ($CPP_{SMM}$), yield ($Yield_{DMDS}$), and purity of the DMDS ($Purity_{DMDS}$) product were investigated to evaluate the catalytic performance of AC-CoPcS. The new supported catalyst exhibits better catalytic performance than the commercial one and can be properly reused four times to obtain $CPP_{SMM}$ and $Yield_{DMDS}$ higher than 90% and 70%. Under the optimum experimental conditions, the $CPP_{SMM}$ and $Yield_{DMDS}$ could reach as high as 98.7% and 86.8%, respectively, and the purity of the DMDS product is as high as 99.8%. This new supported catalyst exhibits good industrial application prospects.

**Keywords:** Merox process; dimethyl disulfide; metal phthalocyanine; activated carbon; sodium methylmercaptide oxidation

---

## 1. Introduction

As an important fine chemical product, dimethyl disulfide (DMDS) is widely used as a soil fumigant [1–3], herbicide [4], food additive [5], sulfide hydrogenation catalyst, and chemical raw material [6]. Thus, the preparation method of DMDS is vital for all the chemical companies, which is commonly classified as four kinds of manufacturing processes: (1) the dimethyl sulfate process

(Equation (1)) [7]; (2) the methanol vulcanization process (Equation (2)) [8,9]; (3) the methanethiol vulcanization process (Equation (3)) [10,11]; and (4) the methyl mercaptan oxidation process which is also called Merox process (Equation (4)) [12]:

$$(CH_3O)_2SO_2 + Na_2S + S \rightarrow CH_3SSCH_3 + Na_2SO_4 \tag{1}$$

$$2CH_3OH + S + H_2S \overset{catalyst}{\rightarrow} CH_3SSCH_3 + 2H_2O \tag{2}$$

$$2CH_3SH + S \overset{catalyst}{\rightarrow} CH_3SSCH_3 + H_2S \tag{3}$$

$$4CH_3SH + O_2 \overset{catalyst}{\rightarrow} 2CH_3SSCH_3 + 2H_2O \tag{4}$$

Compared with other processes, the Merox process shows many advantages, such as high yield and purity of DMDS products, no usage of highly toxic raw materials, etc. However, the traditional Merox process is a gas-gas reaction, so it has a high risk of explosion and pollutes the environment because of the exhaust gas produced in the process. Additionally, the process has very high requirements for reactor quality, airtightness, and pressure endurance. In order to solve these problems, engineers refined the Merox process of the gas-gas reaction into a gas-liquid reaction using sodium methylmercaptide (SMM) as the reagent, as shown in Equations (5) and (6) [13–15]:

$$CH_3SH + NaOH \rightarrow CH_3SNa + H_2O \tag{5}$$

$$4CH_3SNa + O_2 + 2H_2O \overset{catalyst}{\rightarrow} 2CH_3SSCH_3 + 4NaOH \tag{6}$$

Unlike the catalysts used in the reaction shown in Equation (4) of MgO or $Na_2O$ [12], the commonly-used catalysts in the reaction shown in Equation (6) are metal phthalocyanines and their derivatives, such as sulfonated cobalt phthalocyanine (CoPcS) [13,14], cobalt tetraaminophthalocyanine (CoTAPc) [16], etc. The former is the most widely-used catalyst because of its lower cost, high efficiency, and high selectivity of the catalytic performance. However, the free CoPcS catalyst is hydrosoluble, therefore, it very easily gets into the DMDS product, creating separation problems between the catalyst and the product. This leads to high cost of post-treatment of DMDS products, and the catalyst will not be able to be reused. In order to solve these problems, the researchers supported the metal phthalocyanines on solid supports, such as activated carbon (AC) [17,18], activated carbon fiber [19,20], viscose fibers [21], carbon nanotubes [22], graphene oxide [23], silicas [24], mineral carriers [25,26], $TiO_2$ [27,28], molecular sieves [29], etc. In contrast, AC is the best choice for industrial applications because of its special advantages of high surface area, well-developed pore structure, high absorptivity, and stability. However, the AC-supported sulfonated cobalt phthalocyanine (AC-CoPcS) catalyst is commonly prepared by the physical method of dipping [13,14,17] or dispersion [18]. There still exists the problem of loss of the active ingredient of the supported catalyst, resulting in dissatisfactory reuse performance. Thus, the CoPcS should be supported on AC by the chemical grafting method.

In this paper, we developed a proprietary technology of supported catalyst preparation method through a chemical linkage of ethylenediamine between AC and CoPc. Then the new AC-CoPcS catalyst was characterized by UV−VIS spectroscopy (UV–VIS), Fourier transformation infrared spectrum (FT-IR), BET surface area test, and X-ray photoelectron spectroscopy (XPS) before it was applied to prepare DMDS. During the catalytic experiments, the effect of operation parameters of reaction time ($t$), catalyst dosage ($C_{ca}$), reaction temperature ($T_{re}$), and oxygen pressure ($P(O_2)$) on SMM conversion per pass ($CPP_{SMM}$), yield ($Yield_{DMDS}$), and purity of the DMDS product ($Purity_{DMDS}$) were investigated. Moreover, the catalytic performance comparison between the newly prepared AC-CoPcS and commercial AC-CoPcS was also analyzed.

## 2. Materials and Methods

### 2.1. Materials

The SMM solution (18.0 wt%, free alkali 2.4 wt%), commercial CoPcS (99 wt%, Co $\geq$ 6 wt%), AC-CoPcS (CoPcS $\geq$ 5 wt%, BET 265.4 m$^2$/g, $D_{50}$ = 55.5 μm, prepared by the dipping method) were provided by Chongqing Unis Chemical Co., Ltd. (Chongqing, China). The compressed oxygen (99.99 wt%, 45 kg) and AC (analytical purity, BET 704.8 m$^2$/g, $D_{50}$ = 50.7 μm) were purchased from a local chemical reagent company (Chongqing Maoye Chemical Reagent Co., Ltd., Chongqing, China). The sulfoxide chloride (SOCl$_2$), ethylenediamine (EDA), N, N-dimethylformamide (DMF), absolute ethyl alcohol (CH$_3$CH$_2$OH), and nitric acid (HNO$_3$) are of analytical purity without further purification; methenyl trichloride (CHCl$_3$, HPLC grade); high purity helium (He, 99.999%) and all these experimental chemicals were purchased from Chengdu Aikeda Chemical Reagent Co., Ltd. (Chengdu, Sichuan, China).

### 2.2. Synthesis of AC-Supported CoPcS (AC-CoPcS)

The synthetic procedure of AC-CoPcS has applied for a Chinese patent [30], and mainly includes the following three steps:

#### 2.2.1. Synthesis of Modified AC by Ethylene Diamine (AC-E)

Activated carbon (AC) was washed with deionized water and dried to have 10.0 g of AC which was added to a 250 mL beaker that contains 150 mL nitric acid solution ($V$(HNO$_3$):$V$(H$_2$O) = 1:2). The reaction mixture was stirred at 60 °C for 24 h. The product was then washed with deionized water to neutralize and dried in vacuum oven at 105 °C to obtain nitric acid-treated AC (AC-T). Then 2.0 g AC-T and 5 mL SOCl$_2$ were added into a flask. The reaction mixture was heated at 80 °C for 12 h. After the reaction, the mixture was heated up to 130 °C to evaporate residual SOCl$_2$ and cooled down to room temperature (25 °C). The product was added to 100 mL DMF which dissolved 5 mL EDA and reacted at 50 °C for 8 h. After cooling to room temperature, the product was washed with deionized water, DMF, and absolute ethyl alcohol, respectively, several times, and then dried in a vacuum oven at 105 °C to obtain AC-E. The synthetic process is illustrated in Scheme 1.

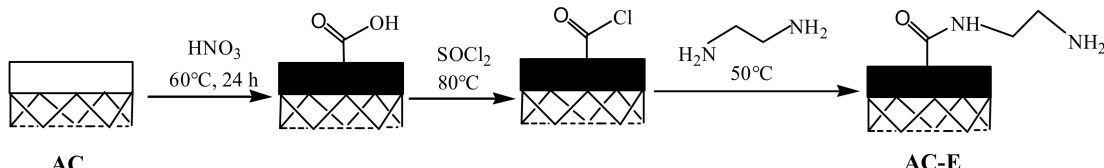

**Scheme 1.** Schematic representation of the preparation of modified AC by ethylene diamine (AC-E).

#### 2.2.2. Synthesis of Modified CoPcS by Sulfoxide Chloride (CoPc(SO$_2$Cl)$_4$)

As CoPcS is a very common chemical product that can be purchased easily, so we do not synthesize it, but purchase it directly. Commercial CoPcS (0.10 g) was dissolved with 100 mL DMF in a flask, and five milliliters of SOCl$_2$ was added to the flask. Then the mixture was heated at 75 °C for 24 h. After the reaction, the product was cooled down to 50 °C and the residual unreacted SOCl$_2$ was removed through vacuum distillation to obtain the CoPc(SO$_2$Cl)$_4$ solution. The synthetic route is illustrated in Scheme 2.

**Scheme 2.** Schematic representation of preparation of modified sulfonated cobalt phthalocyanine by sulfoxide chloride (CoPc(SO$_2$Cl)$_4$).

### 2.2.3. Synthesis of the AC-Supported CoPcS (AC-CoPcS)

Then 2.00 g of previously prepared AC-E was added into the CoPc(SO$_2$Cl)$_4$ solution to react at 45 °C for 12 h and then cooled to room temperature after the reaction. The product was washed by deionized water several times to remove the unreacted component and was dried in a vacuum oven at 60 °C to obtain the final product of AC-CoPcS. The synthetic route is shown in Scheme 3.

**Scheme 3.** Schematic representation showing the preparation of AC-supported sulfonated cobalt phthalocyanine (AC-CoPcS).

### 2.3. Characterization Methods

The UV–VIS absorption spectra were recorded with a UV−VIS absorption spectrometer (U3010, Hitachi, Loveland, CO, USA) by dissolving the CoPcS into DMF with the concentration of $1.0 \times 10^{-6}$ mol/L. The FT-IR spectrum was characterized by a Spectrum One infrared spectrometer (Perkin Elmer, Waltham, MA, USA) using a KBr presser. X-ray photoelectron spectroscopy (XPS) was obtained on an AXIS Ultra DLD Thermo scientific spectroscope (Shimadzu, Kyoto, Japan) with the measurement condition of an X-ray source, a monochromatic aluminum target at 1486.6 eV, ion gun current of 3 mA, and electric voltage of 15 kV. The specific surface area based on nitrogen physisorption of the catalyst samples was measured by a JW-BK100C surface area instrument (JWGB, Sci.&Tech., Beijing, China).

### 2.4. Catalytic Experiment of AC-CoPcS Catalyst for the Preparation of DMDS

The catalytic experiments were carried out in a high-pressure reactor (HPR, $Volume_{max}$ = 1 L, $Pressure_{max}$ = 14.5 MPa, its structure is shown in Figure 1). All the catalytic experiments were carried out in a batch operation, and during each experimental circle, different amounts of AC-CoPcS catalyst was added into the reactor with 450 g SMM solution (18.0 wt%). Before the reaction, the air in the HPR should be replaced by oxygen. Then the control box of the HPR was turned on, and the stirring speed was fixed at 600 rpm. When the temperature in the reactor reached the desired value, the oxygen was inlet to start the catalytic reaction, which was labeled as the zero time point, and the reaction time was recorded accurately. After the reaction, heating electricity was turned off while cooling water was turned on until the temperature in HPR reached room temperature. Before opening the HPR, the exhaust valve was opened to get atmospheric pressure. The oily organic product of DMDS was separated by a separating funnel and was measured by an electronic balance (0.0001 g) to calculate the $Yield_{DMDS}$. The initial and final concentration of SMM solution was determined by the iodometric method to calculate the $CPP_{SMM}$. The principle of the iodometric method is that SMM was reacted by excessive iodine standard solution, and residual iodine was detected by sodium hyposulfite standard solution. The AC-CoPcS catalyst in the aqueous phase was filtered, washed by deionized water, and dried in a vacuum oven for reuse. All the catalytic experiments were conducted in triplicate and the statistical variance was found not to exceed five percent.

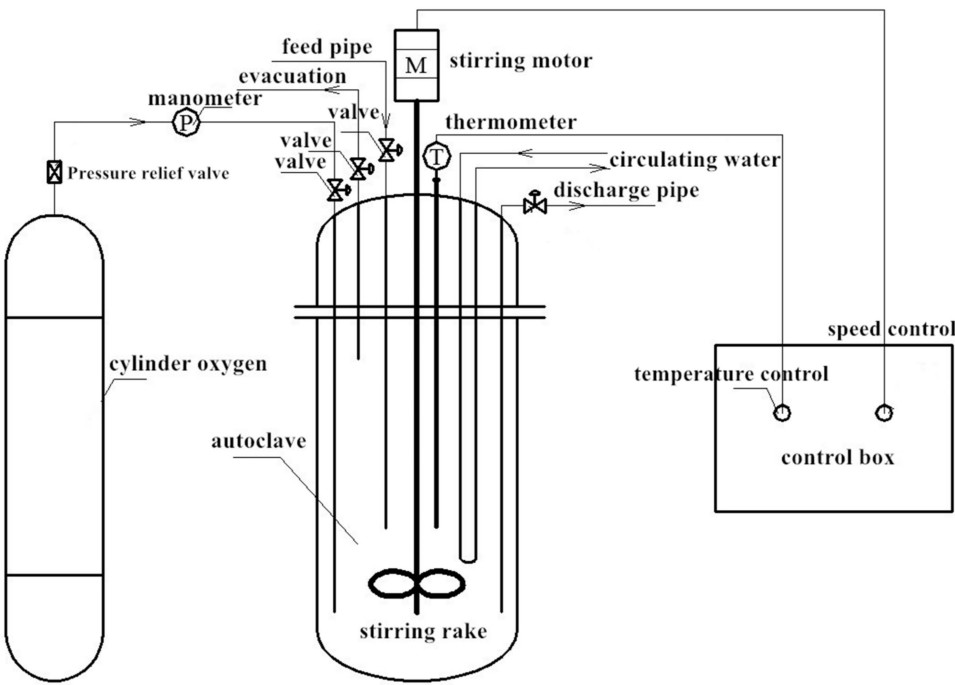

**Figure 1.** Structure schematic diagram of the reactor and reaction process for the preparation of DMDS.

### 2.5. The Determination of Purity of DMDS

The purity of DMDS was calculated by the percentage of different components of the organic product which was measured by a GC6890/MS5973 gas Chromatograph-Mass Spectrometer (GC-MS, Agilent, Santa Clara, CA, USA) with an OV1701 capillary column (30 m × 0.25 mm × 0.25 μm). During each GC-MS test, a test sample of DMDS was prepared by dissolving 1.00 g of DMDS product in 100 mL of methenyl trichloride, and the sample injection volume is 0.5 μL. The heating program of the GC-MS is 50 °C kept for 3 min, then heated at a rate of 10 °C/min to 150 °C and kept for 3 min.

### 2.6. Calculate Method of CPP<sub>SMM</sub> and Yield<sub>DMDS</sub>

*2.6. Calculate Method of $CPP_{SMM}$ and $Yield_{DMDS}$*

The conversion per pass of SMM and the yield of the DMDS were calculated as shown in the following equations:

$$CPP_{SMM} = \frac{consumed\ mol\ of\ SMM\ in\ the\ reaction}{nitial\ mol\ of\ SMM} \tag{7}$$

$$Yield_{DMDS} = \frac{quality\ of\ generated\ DMDS\ in\ the\ reaction}{nitial\ quality\ of\ the\ SMM} \tag{8}$$

## 3. Results and Discussion

### 3.1. Characterization of CoPcS and AC-CoPcS Catalysts

The UV−VIS and FT-IR were used to confirm the structure of the purchased CoPcS catalyst. The metal phthalocyanines and their derivatives have special absorption spectra in the ultraviolet and visible regions. That is the Q band of 600~800 nm and B band or Sorct band of 300–400 nm [31]. The UV−VIS spectra result of CoPcS catalyst is shown in Figure 2. The CoPcS have obvious absorption spectra in the B band and Q band with the maximum wavelengths located at 326 nm and 664 nm, which can be confirmed as the characteristic absorption peak of the CoPcS [32].

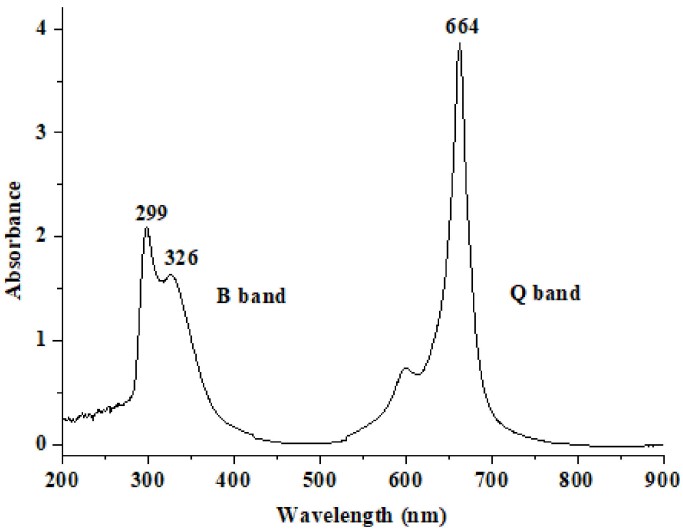

**Figure 2.** UV–VIS spectra of the free CoPcS catalyst (test sample of CoPcS DMF solution with the concentration of $1.0 \times 10^{-6}$ mol/L).

FT-IR result of CoPcS is shown in Figure 3, the peaks of 1716, 1542, 1510, and 1400 cm$^{-1}$ corresponded to C=N stretching vibration, C–H bending vibration, C=C skeleton vibration in the benzene ring, and C-C stretching, respectively. We also found the peaks of 1120, 1053, 914, 732, and 717 cm$^{-1}$ of CoPc reported by Seoudi et al. [33], and two characteristic peaks at 623 and 484 cm$^{-1}$ owing to the phthalocyanine skeletal and cobalt ligand vibrations reported by Zhang et al. [34] confirming the structure of the CoPcS.

XPS was used to analyze the surface element and its chemical states of the AC-CoPcS catalyst, and the results are shown in Figure 4. The surface element of the AC-CoPcS is mainly carbon (C), oxygen (O), nitrogen (N), cobalt (Co,) and sulfur (S). This result indicates the fact that CoPcS was successfully combined into the AC through a chemical linkage of ethylenediamine because the Co and S appear in the XPS pattern of the new material. The CoPcS is very easy to dissolve in water, while the AC-CoPcS catalyst is well washed by deionized water after being prepared, so CoPcS appearing in the new materials cannot be physically adsorbed by the AC, but is chemically connected with it.

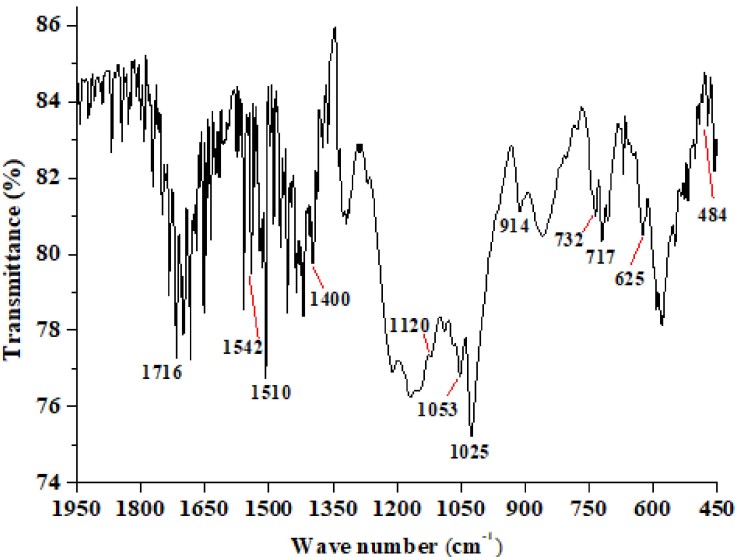

**Figure 3.** The FT-IR spectra of the free CoPcS catalyst (KBr tabletting treatment).

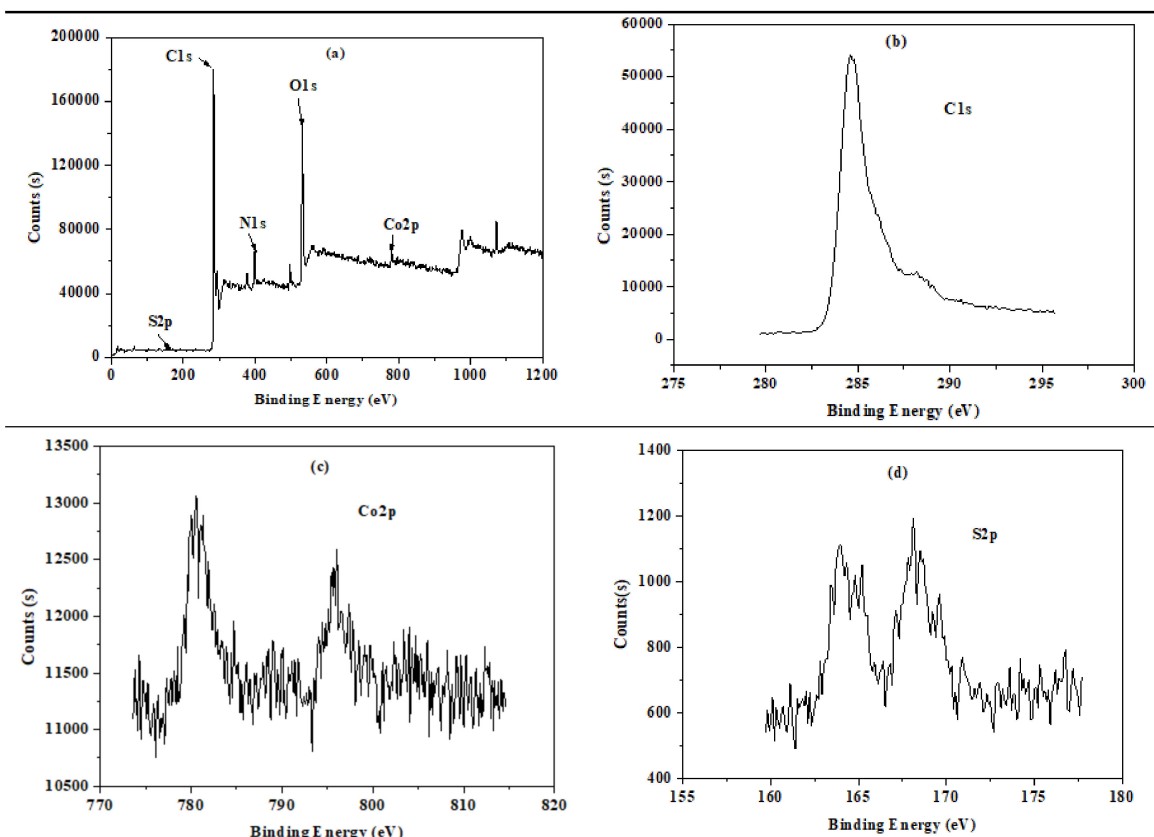

**Figure 4.** The XPS spectra of the AC-CoPcS (**a**), C1s (**b**), Co2p (**c**), and S2p (**d**).

The surface area results of the AC and AC-supported catalysts are listed in Table 1, the BET specific surface area of AC support, commercial, and newly prepared AC-CoPcS catalysts are 704.8 $m^2$/g, 265.4 $m^2$/g, and 301.3 $m^2$/g. Compared with the commercial AC-CoPcS, the new AC-CoPcS catalyst shows a slightly higher BET specific surface area, larger total pore volume, and smaller average pore size, which may benefit the adsorption of the reagents to accelerate the catalytic reaction, but this should be proved by the catalytic performance of the two catalysts.

**Table 1.** The surface area test results of the AC, commercial, and newly prepared AC-CoPcS catalysts.

| Sample | BET Surface Area ($m^2/g$) | Total Pore Volume ($cm^3/g$) | Average Pore Size (nm) |
|---|---|---|---|
| AC | 704.8 | 1.30 | 2.50 |
| AC-CoPcS (commercial) | 265.4 | 0.46 | 4.16 |
| AC-CoPcS (new) | 301.3 | 0.51 | 3.95 |

*3.2. The Effect of Operation Parameters on AC-CoPcS Catalytic Performance*

3.2.1. The Performance Comparison between Free CoPcS and AC-CoPcS Catalysts

In order to show the advantages of the supported catalyst, the catalytic performance of free CoPcS and AC-CoPcS catalysts were compared in Table 2 and Figure 5. As is shown in Figure 5, the free CoPcS catalyst is so highly soluble that it runs into not only the water layer but also the organic layer of DMDS product, resulting in post-treatment of DMDS by AC absorption and an inability to reuse the catalyst. However, the supported AC-CoPcS catalyst could be a good solution to these problems: as is shown in Figure 5b, the AC-CoPcS catalyst only stays in the water layer and can be reused through simple filtration and washing treatment. In addition, the AC-CoPcS also exhibits better catalytic performance than the free CoPcS catalyst: as is shown in Table 2, the *Yield*$_{DMDS}$ and *Purity*$_{DMDS}$ using the AC-CoPcS catalyst are higher than the one using the free CoPcS catalyst of 86.8% to 81.4% and 99.8% to 98.7%, respectively. The better catalytic performance of the AC-CoPcS catalyst could be attributed to the following reasons: (1) Large specific surface area and flourishing pore structure of AC support, which can not only absorb the reagents of SMM and oxygen to react on its surface, but also remove the byproduct of the reaction to improve the yield and purity of DMDS; or (2) AC itself can be used as the catalyst in the Merox process, which can also help to improve the catalytic performance of the new AC-CoPcS catalyst [35].

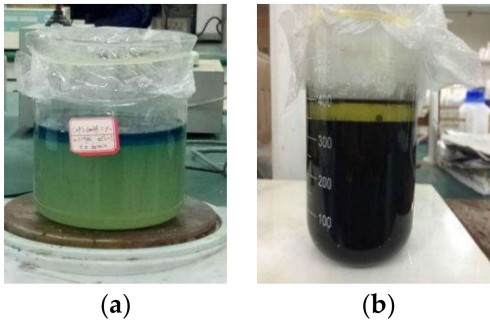

|                  |                  |
| :--------------: | :--------------: |
|       (a)        |       (b)        |

**Figure 5.** The unseparated DMDS product (upper layer) after the catalytic oxidation reaction using (**a**) free CoPcS and (**b**) AC-CoPcS catalysts. Experimental conditions: SMM (18.0 wt%) 450 g, $RR$ = 600 r/min, free CoPcS dosage 160 ppm, AC-CoPcS dosage 888.9, $t$ = 60 min, $T_{re}$ = 65 °C, $P(O_2)$ = 0.9 MPa.

**Table 2.** The performance comparison between free CoPcS and AC-CoPcS catalysts [1].

| Catalysts | $C_{ca}$ | *Yield*$_{DMDS}$ | *Purity*$_{DMDS}$ | Post-Treatment of DMDS |
|---|---|---|---|---|
| CoPcS | 160 ppm | 81.4% | 98.7% | AC adsorption |
| AC-CoPcS | 888.9 ppm | 86.8% | 99.8% | None |

[1] Other experimental conditions: SMM (18.0 wt%) 450 g, $RR$ = 600 r/min, $t$ = 60 min, $T_{re}$ = 65 °C, $P(O_2)$ = 0.9 MPa.

3.2.2. The Effect of Reaction Time ($t$)

The reaction time is one of the most important process parameters that should be optimized firstly, so the effect of reaction time on $CPP_{SMM}$, *Yield*$_{DMDS}$, and *Purity*$_{DMDS}$ was investigated. The purity

of the DMDS product was detected by GC-MS, and the spectra are shown in Figure 6. As can be seen, the retention time of DMDS is around 4 min, and there is no other peak of the byproduct in the GS-MS spectra, indicating the high quality of the DMDS is prepared by the new catalytic reaction system. As is shown in Figure 7, when the reaction time increases from 30 min to 60 min, the $CPP_{SMM}$ and $Yield_{DMDS}$ increase gradually from 62.4% and 44.1% to 98.7% and 86.8%, respectively. However, when the reaction time continues to increase from 60 min to 90 min, the $CPP_{SMM}$ decreases slightly to 84.1% at first and then remains nearly constant, while the $Yield_{DMDS}$ remains constant at the same time. Nevertheless, the reaction time seems have a little effect on the $Purity_{DMDS}$, when the reaction time increases from 30 min to 90 min, the $Purity_{DMDS}$ decreases slightly from 99.8% to 99.1%, which verifies the high quality of the DMDS product that was prepared in the new system. As a result, the reaction time of 60 min is optimal, when the reaction time extends to 75 min or 90 min, the DMDS product may react with each other to form the byproduct, causing the decrease of $Yield_{DMDS}$ and $Purity_{DMDS}$ [30].

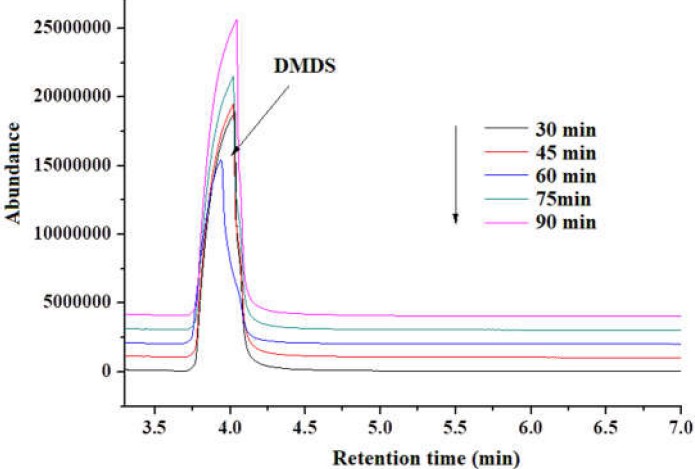

**Figure 6.** The GC-MS spectra of the DMDS product under different reaction time. Experimental conditions: SMM (18.0 wt%) 450 g, $RR$ = 600 r/min, $C_{ca}$ = 888.9 ppm, $T_{re}$ = 65 °C, $P(O_2)$ = 0.9 MPa.

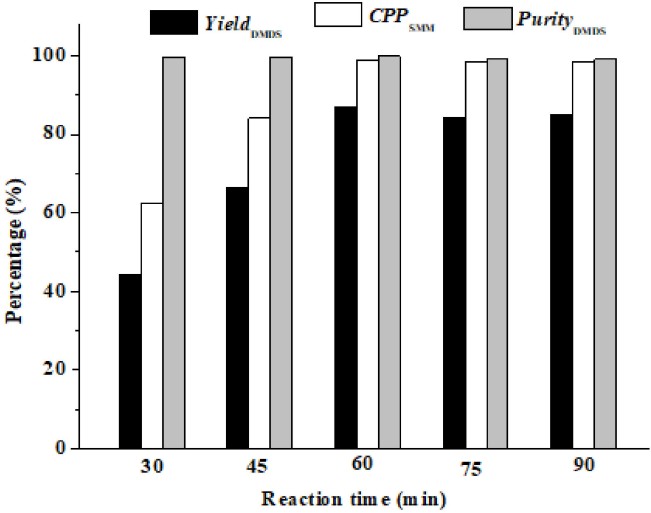

**Figure 7.** The effect of reaction time on $CPP_{SMM}$, $Yield_{DMDS}$, and $Purity_{DMDS}$ using the AC-CoPcS catalyst. Experimental conditions: SMM (18.0 wt%) 450 g, $RR$ = 600 r/min, $C_{ca}$ = 888.9 ppm, $T_{re}$ = 65 °C, $P(O_2)$ = 0.9 MPa.

### 3.2.3. The Effect of AC-CoPcS Catalyst Dosage ($C_{ca}$)

Figure 8 shows the effect of AC-CoPcS catalyst dosage ($C_{ca}$) on $CPP_{SMM}$, $Yield_{DMDS}$, and $Purity_{DMDS}$. The $CPP_{SMM}$ and $Yield_{DMDS}$ increase gradually when the $C_{ca}$ increases from 222.2 ppm

(µg/g, 0.1 g/450 g) to 888.9 ppm and then begins to decrease continuously when $C_{ca}$ continues to increase to 1555.6 ppm. The optimum AC-CoPcS catalyst dosage is 888.9 ppm where maximum $CPP_{SMM}$ and $Yield_{DMDS}$ of 98.7% and 86.8% are achieved. When the $C_{ca}$ increases, the more active spot of the catalyst and reaction surface involves, so the reaction rate of Equation (6) process accelerates, resulting in a sharp increase of the $CPP_{SMM}$ and $Yield_{DMDS}$ at first. However, when it reaches, and even exceeds, the appropriate dosage, too much heterogeneous solid matter may hinder the gas-liquid mass transfer between SMM and oxygen, so the $CPP_{SMM}$ and $Yield_{DMDS}$ decrease gradually. However, $C_{ca}$ seems to have no effect on the purity of the DMDS product (Figure 8), and the $Purity_{DMDS}$ is nearly constant and its value of all the samples is above 99.5%.

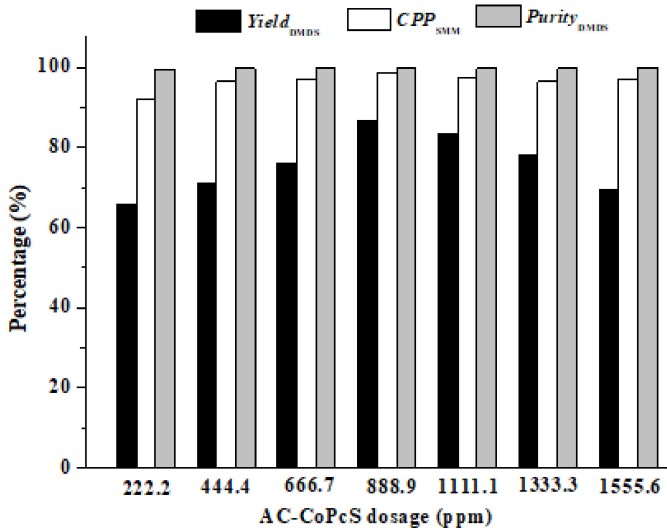

**Figure 8.** The effect of AC-CoPcS catalyst dosage on $CPP_{SMM}$, $Yield_{DMDS}$, and $Purity_{DMDS}$. Experimental conditions: SMM (18.0 wt%) 450 g, $RR$ = 600 r/min, $t$ = 60 min, $T_{re}$ = 65 °C, $P(O_2)$ = 0.9 MPa.

### 3.2.4. The Effect of Reaction Temperature ($T_{re}$)

Commonly, the reaction temperature has a great influence on the reaction rate. The effect of $T_{re}$ on the $CPP_{SMM}$, $Yield_{DMDS}$, and $Purity_{DMDS}$ was investigated when other operation parameters kept constant, and the result is shown in Figure 9. The $CPP_{SMM}$ and $Yield_{DMDS}$ increase gradually when $T_{re}$ increases from 35 °C to 65 °C, but it decreases slightly when $T_{re}$ continues to increase from 65 °C to 75 °C, so the optimum $T_{re}$ is 65 °C where the best $CPP_{SMM}$ and $Yield_{DMDS}$ are achieved. The $T_{re}$ seems to have little effect on $Purity_{DMDS}$ when it is lower than 65 °C, however, when $T_{re}$ increases from 65 °C to 75 °C the $Purity_{DMDS}$ decreases sharply from 99.8% to 98.0%. The reason for these results is that the reaction of the SMM and oxygen is endothermal, and the reagents are easier to activate, which accelerates the reaction rate when $T_{re}$ increases, leading to the increase of the $CPP_{SMM}$ and $Yield_{DMDS}$. However, extremely higher $T_{re}$ of 75 °C brings not only the byproduct formation problems, which will seriously affect the purity of DMDS product, but also the decrease of $Yield_{DMDS}$ because of the volatilization of DMDS into the gaseous state [6]. When $T_{re}$ is at 75 °C, a white cotton-like byproduct in the organic layer was observed in the experiment.

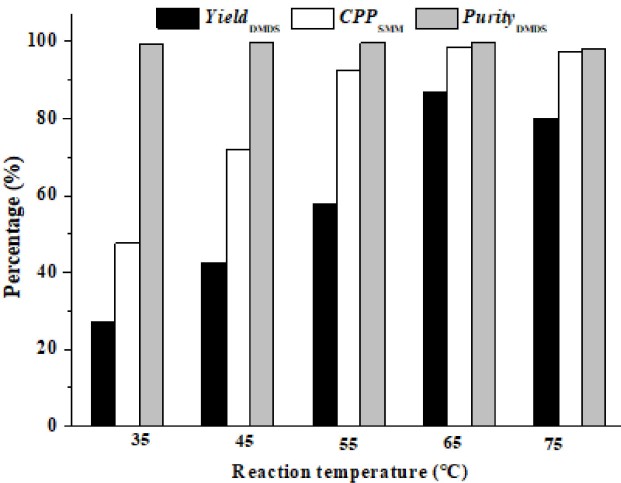

**Figure 9.** The effect of reaction temperature on $CPP_{SMM}$, $Yield_{DMDS}$, and $Purity_{DMDS}$ using the AC-CoPcS catalyst. Experimental conditions: SMM (18.0 wt%) 450 g, $RR$ = 600 r/min, $C_{ca}$ = 888.9 ppm, $t$ = 60 min, $P(O_2)$ = 0.9 MPa.

### 3.2.5. The Effect of Oxygen Pressure ($P(O_2)$)

As is shown in Equation (6), the synthesis reaction of DMDS is a volume reduction reaction, so the increase of $P(O_2)$ will benefit the process. The effect of $P(O_2)$ on the $CPP_{SMM}$, $Yield_{DMDS}$, and $Purity_{DMDS}$ was investigated, and the result is shown in Figure 10. As is shown in the figure, the $Yield_{DMDS}$ increases sharply from 42.5% to 86.8% with the $P(O_2)$ increasing from 0.5 MPa to 0.9 MPa, and it decreases slightly at 1.0 MPa. However, at the same time, the $CPP_{SMM}$ only increases from 90.5% to 98.7% when the $P(O_2)$ increases from 0.5 MPa to 0.9 MPa, and then also decreases slightly to 96.8% when $P(O_2)$ continues to increase to 1.0 MPa. These results indicate the fact that SMM is much easier to react with oxygen, but the reaction condition is much harsher to obtain a high yield of the DMDS product. Nevertheless, the purity of the DMDS was not affected by the $P(O_2)$, and the $Purity_{DMDS}$ is still as high, above 99.5%.

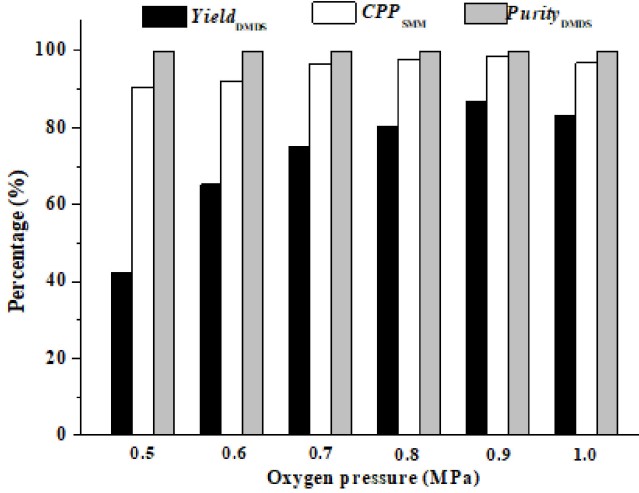

**Figure 10.** The effect of oxygen pressure on $CPP_{SMM}$, $Yield_{DMDS}$, and $Purity_{DMDS}$ using the AC-CoPcS catalyst. Experimental conditions: SMM (18.0 wt%) 450 g, $RR$ = 600 r/min, $C_{ca}$ = 888.9 ppm, $t$ = 60 min, $T_{re}$ = 65 °C.

### 3.3. The Performance Comparison between Commercial and Newly Prepared AC-CoPcS Catalysts

The same optimization method of the operation parameters was applied to the commercial AC-CoPcS catalyst, and the optimum parameters are listed in Table 3. The best catalyst dosage of

commercial AC-CoPcS of 1333.3 ppm is much larger than the newly prepared one of 888.9 ppm, which will cost more for industrial application. The performance comparison experiment between the commercial and new AC-CoPcS catalysts was conducted under their optimum parameters, and the result is summarized in Table 3. When compared with commercial AC-CoPcS, the newly prepared AC-CoPcS catalyst shows higher $CPP_{SMM}$ and $Yield_{DMDS}$ of 98.7% and 86.8% than the commercial one of 97.4% and 85.6%. However, the purity of the DMDS product using both the commercial and new AC-CoPcS catalysts are particularly high, 99.7% and 99.8%, respectively.

The most important advantage of the supported catalyst is its reusability. The reuse performance of the commercial and newly prepared AC-CoPcS catalyst is shown in Figures 11 and 12. As can be seen in the figures, $CPP_{SMM}$ and $Yield_{DMDS}$ decrease with the increase of reuse times of the two catalysts. For the new AC-CoPcS catalyst, it can be reused seven times, and for the first four times of reuse the $CPP_{SMM}$ and $Yield_{DMDS}$ only reduce from 98.7% and 86.8% to 92.0% and 72.7%, respectively. However, the significant decrease of the $CPP_{SMM}$ and $Yield_{DMDS}$ occurs at the fifth reuse of the new catalyst of 78.2% and 46.6%, and the $CPP_{SMM}$ and $Yield_{DMDS}$ is only 26.2% and 8% for the seventh reuse. The reuse performance of the commercial AC-CoPcS catalyst is slightly worse than the former one. It can only be reused five times. The $CPP_{SMM}$ and $Yield_{DMDS}$ decrease from 97.4% and 85.6% to 90.2% and 71.3% for the second reuse, but decrease to 70.8% and 33.9% for the third reuse. As a result, the new and commercial AC-CoPcS catalysts can be properly reused four and two times, respectively, which still can achieve relatively high $Yield_{DMDS}$ and $CPP_{SMM}$ of more than 70% and 90%. It is worth mentioning that the reuse times have little effect on the $Purity_{DMDS}$. The $Purity_{DMDS}$ of all the DMDS product remains above 99.3%. The decrease of catalytic performance of AC-CoPcS catalyst with the increase of reuse times can be attributed to the following reasons: Firstly, the more times AC-CoPcS catalyst is reused, the greater loss of its active components during the reaction process of high temperature and high pressure; secondly, the SMM reagent, DMDS product, or even the byproduct produced in the reaction may be absorbed by the AC carrier, which can block up the micropores of the AC, preventing the reagents and catalyst from contacting with each other; and, thirdly, the mechanical strength of the AC carrier will also drop, resulting in its hole collapse, which also reduces the connection of active spots of the catalyst with the reagents [30].

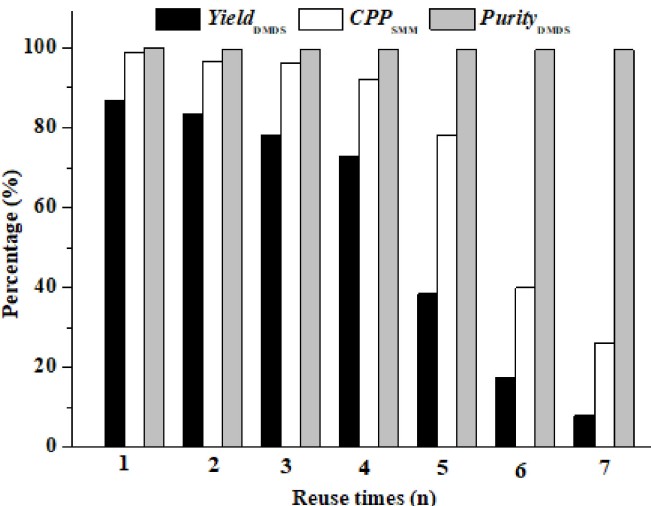

**Figure 11.** The effect of reuse times of new AC-CoPcS catalyst on $CPP_{SMM}$, $Yield_{DMDS}$, and $Purity_{DMDS}$. Experimental conditions: SMM (18.0 wt%) 450 g, $RR$ = 600 r/min, $C_{ca}$ = 888.9 ppm, $t$ = 60 min, $T_{re}$ = 65 °C, $P(O_2)$ = 0.9 MPa.

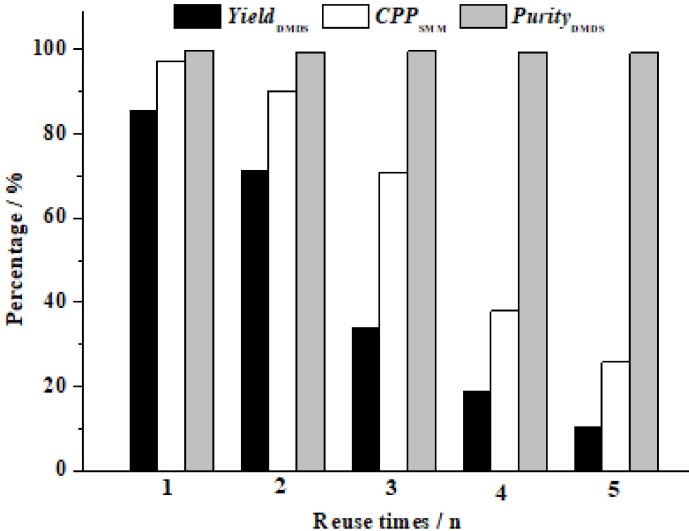

**Figure 12.** The effect of reuse times of commercial AC-CoPcS catalyst on $CPP_{SMM}$, $Yield_{DMDS}$, and $Purity_{DMDS}$. Experimental conditions: SMM (18.0 wt%) 450 g, $RR$ = 600 r/min, $C_{ca}$ = 1333.3 ppm, $t$ = 60 min, $T_{re}$ = 65 °C, $P(O_2)$ = 0.9 MPa.

**Table 3.** The performance comparison between commercial and newly-prepared AC-CoPcS catalysts.

| Catalyst | Synthetic Method | $CPP_{SMM}$ | $Yield_{DMDS}$ | $Purity_{DMDS}$ | Optimum Operation Parameters | | | | Proper Reuse Times |
| --- | --- | --- | --- | --- | --- | --- | --- | --- | --- |
| | | | | | $t$ | $C_{ca}$ | $T_{re}$ | $P(O_2)$ | |
| **AC-CoPcS (new)** | chemical grafting | 98.70% | 86.8% | 99.8% | 60 min | 888.9 ppm | 65 °C | 0.9 MPa | 4 |
| **AC-CoPcS (commercial)** | physical dipping | 97.4% | 85.6% | 99.7% | 60 min | 1333.3 ppm | 65 °C | 0.9 Mpa | 2 |

## 4. Conclusions

A new activated carbon (AC)-supported sulfonated cobalt phthalocyanine (AC-CoPcS) catalyst for a refined Merox process of SMM oxidation for the preparation of DMDS was successfully synthesized by the chemical grafting method. The new AC-CoPcS catalyst does not get into the DMDS product and can be easily separated for reuse at least four times to keep $CPP_{SMM}$ and $Yield_{DMDS}$ higher than 90% and 70%. The operation parameters of the catalytic reaction were optimized, and the optimum conditions for the new reaction system are a reaction time of 60 min, AC-CoPcS dosage of 888.9 ppm, reaction temperature of 65 °C, and oxygen pressure of 0.9 MPa. Under the optimum conditions, the maximum $CPP_{SMM}$ and $Yield_{DMDS}$ are 98.7% and 86.8%. The purity of the DMDS product can reach as high as 99.8%. Compared with commercial AC-CoPcS prepared by the physical dipping method, the newly prepared AC-CoPcS catalyst shows a better catalytic performance with higher $CPP_{SMM}$ and $Yield_{DMDS}$. Additionally, it can be properly reused twice as often as the commercial one but only two-thirds of its catalyst dosage is needed.

## 5. Patents

The refined Merox process of Equations (5) and (6) for the preparation of DMDS is our patented technology (CN 105924372 B), and the synthetic method of the supported AC-CoPcS catalyst has already applied for a Chinese patent (CN 106984361 A). No person or organization is allowed to use these technologies for commercial purposes unless admitted by Chongqing Unis Chemical Company Ltd. (CUC).

**Author Contributions:** X.Q. and S.L. designed the work and revised the initial draft of the paper; M.D. and Z.C. conducted the experiment; Z.C. wrote the initial draft of the paper; and D.Z., Y.L., and R.Y. discussed data analysis and gave many constructive suggestions.

**Funding:** This research was funded by the Postdoctoral Research Project of Chongqing (Xm2016063), the Research Project of Chongqing Unis Chemical Company Ltd. (2016Q28), the Foundation and Frontier Research Project of Chongqing (cstc2015jcyjA20005), and the Scientific and Technological Research Program of Chongqing Municipal Education Commission (KJ1600927).

**Conflicts of Interest:** The authors declare no conflict of interest.

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
