# Peer review of "Synthesis and Catalytic Activity of Activated Carbon Supported Sulfonated Cobalt Phthalocyanine in the Preparation of Dimethyl Disulfide"

_applsci, doi:10.3390/app9010124_

Round 1

Reviewer 1 Report

I believe the article “Synthesis and Catalytic Activity of Activated Carbon Supported Sulphonated Cobalt Phthalocyanine in the Preparation of Dimethyl Disulfide” by Cheng et al is suitable for publication in Applied Sciences only after major changes in editing. The subject addressed is here worthy of investigation, and new catalyst exhibits good industrial application prospect. However, the manuscript has a lot of grammatical errors. For the understanding of the readers, the whole manuscript needs to be revised. Thus it can’t be accepted in its present form. 

For instance,

2.2.1. Synthesis of modified AC by ethylene diamine (AC-E)

“Firstly, the deionized water washed and then dried AC (10.0 g) was placed in a 250 mL beaker which contains 150 ml nitric acid solution (V(HNO3) : V(H2O) = 1 : 2) stirred at 60 for 24 h. The product was washed by deionized water to neutral and dried in vacuum oven at 105 to get nitric acid treated AC (AC-T). Then 2.0 g AC-T and 5 mL SOCl2 were added into a flask heated to 80….” Should be written something like

Activated carbon (AC) was washed with the deionized water and dried to have 10.0 g of AC which was added to 250 mL beaker which contains 150 ml nitric acid solution (V(HNO3) : V(H2O) = 1 : 2). The reaction mixture was stirred at 60 for 24 h. The product was then washed with deionized water to neutralize and dried in vacuum oven at 105 to get nitric acid treated AC (AC-T). Then 2.0 g AC-T and 5 mL SOCl2 were added into a flask. The reaction mixture was heated at 80

Author Response

Response to Reviewer 1 Comments

Point 1: The manuscript has a lot of grammatical errors. For the understanding of the readers, the whole manuscript needs to be revised.

Response 1:  As non-speaking English researchers, we are really sorry for the grammatical errors in the manuscript. In order to solve this problem, we invite an English speaker to revise the manuscript for us. The detailed revision is shown as follows and revised in the manuscript accordingly in red.

1、Merox process---------The Merox process

2、sulfiding hydroprocessing catalysts------- sulfide hydrogenation catalyst

3、gas-gas reaction  ------- a gas-gas reaction

4、using sodium methylmercaptide (SMM) as reagent-----using sodium methylmercaptide

(SMM) as the reagent

5、post treatment---- post-treatment

6、adsorptivity--- absorptivity

7、sulphonated cobalt phthalocyanine---- sulfonated cobalt phthalocyanine

8、by physical method of dipping --- by the physical method of dipping 

9、through chemical linkage of ethylenediamine between AC and CoPc--- through a chemical linkage of ethylenediamine between AC and CoPc

10、         CoPcS and commercial AC- CoPcS was also analysed----- CoPcS and commercial AC- CoPcS was also analyzed

11、         ethylene diamine--- ethylenediamine

12、         “Firstly, the deionized water washed and then dried AC (10.0 g) was placed in a 250 mL beaker which contains 150 ml nitric acid solution (V(HNO3) : V(H2O) = 1 : 2) stirred at 60℃ for 24 h. The product was washed by deionized water to neutral and dried in vacuum oven at 105℃ to get nitric acid treated AC (AC-T). Then 2.0 g AC-T and 5 mL SOCl2 were added into a flask heated to 80℃------ Activated carbon (AC) was washed with deionized water and dried to have 10.0 g of AC which was added to a 250 mL beaker that contains 150 ml nitric acid solution (V(HNO3) : V(H2O) = 1 : 2). The reaction mixture was stirred at 60℃ for 24 h. The product was then washed with deionized water to neutralize and dried in vacuum oven at 105℃ to get nitric acid treated AC (AC-T). Then 2.0 g AC-T and 5 mL SOCl2 were added into a flask. The reaction mixture was heated at 80℃

13、         the product was washed by deionized water, DMF and absolute ethyl alcohol respectively for several times---- the product was washed by deionized water, DMF and absolute ethyl alcohol respectively several times

14、         was washed by deionized water---- was washed with deionized water,

15、         so we don’t synthetize but purchase it directly---- so we don’t synthesize but purchase it directly

16、         Five milliliter SOCl2 was placed into a flask with 100 mL DMF dissolved 0.10 g fore-mentioned commercial CoPcS, and then the mixture was heated at 75℃ for 24 h.---- Commercial CoPcS (0.10 g) was dissolved with 100 mL DMF in a flask, and five milliliter SOCl2 was added to the flask and then the mixture was heated at 75℃ for 24 h.

17、         Then 2.00 g previously prepared AC-E was added into the CoPc(SO2Cl)4 solution to react at 45℃ for 12 h and then was cooled to room temperature after reaction.---- Then 2.00 g previously prepared AC-E was added into the CoPc(SO2Cl)4 solution to react at 45℃ for 12 h and then cooled to room temperature after the reaction.

18、         The UV-Vis absorption spectra was recorded with a UV−Vis absorption spectrometer---- The UV-Vis absorption spectra were recorded with a UV−Vis absorption spectrometer

19、         X ray source---- an X-ray source

20、         The oily organic product of DMDS was separated by separating funnel, and was measured by electronic balance---- The oily organic product of DMDS was separated by separating funnel and was measured by electronic balance

21、         washed by deionized newater------ washed by deionized water

22、         The initial and final concentration of SMM solution was determined by iodometric method to calculated the CPPSMM------ The initial and final concentration of SMM solution was determined by the iodometric method to calculate the CPPSMM.

23、         and dried in vacuum oven for reuse--- and dried in a vacuum oven for reuse.

24、         The conversion per pass of SMM and the yield of the DMDS were calculated as the following equation shown---- The conversion per pass of SMM and the yield of the DMDS were calculated as shown in the following equations

25、         This result indicates the fact that CoPcS was successfully combined into the AC through chemical linkage of ethylenediamine----- This result indicates the fact that CoPcS was successfully combined into the AC through a chemical linkage of ethylenediamine

26、         but this should be proved by catalytic performance of the two catalysts---- but this should be proved by the catalytic performance of the two catalysts

27、         commercial and newly-prepared AC-CoPcS catalysts are 704.8 m2/g, 265.4 m2/g and 301.3 m2/g------ commercial and newly-prepared AC-CoPcS catalysts are 704.8 m2/g, 265.4 m2/g, and 301.3 m2/g.

28、         in order to show the advantages of supported catalyst--- in order to show the advantages of the supported catalyst

29、         in water layer--- in the water layer

30、         could be attributed to following reasons:---- could be attributed to the following reasons:

31、         the PurityDMDS deceases a little---- the PurityDMDS decreases a little

32、         The purity of the DMDS product was detected by GC-MS, and the spectra is shown in Figure 6---- The purity of the DMDS product was detected by GC-MS, and the spectra are shown in Figure 6

33、         no other peak of byproduct in the GS-MS spectra--- no other peak of the byproduct in the GS-MS spectra

34、         indicating high quality of the DMDS---- indicating the high quality of the DMDS

35、         As a result, reaction time of 60 min is the optimal--- As a result, the reaction time of 60 min is optimal

36、         As can be seen, the retention time of DMDS is 4.5 min----- As can be seen, the retention time of DMDS is around 4 min.

37、         causing decrease of YieldDMDS and PurityDMDS--- causing the decrease of YieldDMDS and PurityDMDS

38、         resulting in sharp increase of ---- resulting in a sharp increase of the

39、         so the CPPSMM and YieldDMDS decreases gradually---- so the CPPSMM and YieldDMDS decrease gradually.

40、         the purity of DMDS product---- the purity of the DMDS product

41、         white cotton-like by product--- a white cotton-like by-product

42、         The Tre seems have little effect on---- The Tre seems to have little effect on

43、         the volatilization of DMDS into gaseous state--- the volatilization of DMDS into the gaseous state

44、         get high yield of DMDS product---- get a high yield of DMDS product.

45、         the CPPSMM and YieldDMDS only reduces from 98.7% and 86.8% to 92.0% and 72.7% respectively----- the CPPSMM and YieldDMDS only reduce from 98.7% and 86.8% to 92.0% and 72.7% respectively.

46、         However, significant decrease of the CPPSMM and YieldDMDS occurs at----- However, the significant decrease of the CPPSMM and YieldDMDS occurs at

47、         the mechanical strength of the AC carrier will also drops----- the mechanical strength of the AC carrier will also drop

48、         but only two thirds of its catalyst dosage is needed----- but only two-thirds of its catalyst dosage is needed.

49、         The new AC-CoPcS catalyst don’t go into the DMDS product--- The new AC-CoPcS catalyst doesn’t go into the DMDS product

50、         Compared with commercial AC-CoPcS prepared by physical dipping method---- Compared with commercial AC-CoPcS prepared by the physical dipping method

51、         the newly-prepared AC-CoPcS catalyst shows better catalytic performance--- the newly-prepared AC-CoPcS catalyst shows a better catalytic performance

Reviewer 2 Report

In the present article, the authors have described the synthesis and characterization of active carbon (AC) solid supported CoPcS, a well-known catalyst widely used in DMDS synthesis. There have been multiple examples of solid supported CoPcS including AC supported CoPcS in the literature as well as in the market. In this context, the present article does not contain a notable novelty. But the authors have successfully demonstrated that their AC-CoPcS shows equal efficiency in catalytic activity compared to the commercially available version and their new AC-CoPcS catalyst showed double reusability compared to the commercial version. In this context, the present article is important and may appear cost effective in the industrial process, though more studies and optimizations are required to obtain a definitive cost of the new method and that is beyond the scope of the present article. Hence I recommend the present article to be published after the following minor revision:

1.       Line 64-70, the authors have highlighted the need of solid support-catalyst conjugation by the means of chemical bonds (covalent bonds!) rather the physical interactions. But  AC supported chemically linked CoPcS is reported already(Applied Catalysis B: Environmental 2014, 154-155, 36-43,). The authors have cited the reference (Ref 20) but the way line 64-70 have been presented, it appears the authors are the first to introduce the idea. Please mention the reference 20 in context of line 64-70 and justify the need of your work over the published work.

Author Response

Response to Reviewer 2 Comments

Point 1:  Line 64-70, the authors have highlighted the need of solid support-catalyst conjugation by the means of chemical bonds (covalent bonds!) rather the physical interactions. But  AC supported chemically linked CoPcS is reported already (Applied Catalysis B: Environmental 2014, 154-155, 36-43,). The authors have cited the reference (Ref 20) but the way line 64-70 have been presented, it appears the authors are the first to introduce the idea. Please mention the reference 20 in context of line 64-70 and justify the need of your work over the published work.

Response 1:  Thank you for the reviewer’s comments, the AC supported chemically linked CoPcS is not reported yet, and the synthetic method was applied for Chinese patent (CN 106984361 A) because of its novelty. The Ref 20 (Applied Catalysis B: Environmental 2014, 154-155, 36-43) is about cobalt tetraaminophthalocyanine supported onto activated carbon fibers (ACFs-CoPc), which is mainly prepared by a physically dipping method that listed as follows according to the supplementary information of the paper. Besides, the two catalysts are not used in the same area, so it’s difficult to compare their catalytic performance.

Preparation of ACFs-CoPc Catalyst

Activated carbon fibers (ACFs, 40 g) were impregnated into a nitric acid solution (3 M) for 24h at room temperature, and washed with deionized water to neutrality. The dried oxidated ACFs (30 g) were immersed into CoPc aqueous solution with a desired concentration; the reaction was carried out at 60 °C for 1 h, and then it was processed at 70 °C by addition of proper amount of Na2CO3 for 2 h. After another 2 h reaction at 75 °C, the resulting product was dried at 80 °C. The unreacted CoPc was removed by washing with distilled water five times. Finally, the product (ACFs-CoPc) was dried at 80 °C.
